# Colour selective control of terahertz radiation using two-dimensional hybrid organic inorganic lead-trihalide perovskites

Ashish Chanana [1], Yaxin Zhai[2], Sangita Baniya[2], Chuang Zhang [2,3], Z. Valy Vardeny[2] & Ajay Nahata[1]

Controlling and modulating terahertz signals is of fundamental importance to allow systems level applications. We demonstrate an innovative approach for controlling the propagation properties of terahertz (THz) radiation, through use of both the excitation optical wavelength (colour) and intensity. We accomplish this using two-dimensional (2D) layered hybrid tri-halide perovskites that are deposited onto silicon substrates. The absorption properties of these materials in the visible range can be tuned by changing the number of inorganic atomic layers in between the organic cation layers. Optical absorption in 2D perovskites occurs over a broad spectral range above the bandgap, resulting in free carrier generation, as well as over a narrow spectral range near the bandedge due to exciton formation. We find that only the latter contribution gives rise to photo-induced THz absorption. By patterning multiple 2D perovskites with different optical absorption properties onto a single device, we demonstrate both colour selective modulation and focusing of THz radiation. These findings open new directions for creating active THz devices.

[1] Department of Electrical and Computer Engineering, University of Utah, 50 S. Central Campus Drive, Salt Lake City, UT 84112, USA. [2] Department of Physics and Astronomy, University of Utah, 115 S. 1400 East, Salt Lake City, UT 84112, USA. [3]Present address: Institute of Chemistry, Chinese Academy of Sciences, Beijing 100190, China. Correspondence and requests for materials should be addressed to Z.V.V. (email: val@physics.utah.edu) or to A.N. (email: nahata@ece.utah.edu)

Terahertz (THz) technologies hold great promise in the development of next-generation computing and communication systems[1]. Among the many device capabilities needed to create systems level applications, the ability to control and modulate signals is of fundamental importance[2]. Prior demonstrations of active THz modulators have utilized mechanical[3–6], electrical[7–12], and optical[13–19] means for affecting change. All-optical approaches are particularly attractive because they offer the potential for high-speed modulation and switching, and can be fabricated straightforwardly on conventional semiconductor substrates. Compared to electrically controlled devices, where the need for ohmic contact adds complexity, contact-free optical modulators can be designed more simply. In such devices, modulation of THz radiation can be realized when the incident optical control beam has a photon energy above the bandgap, where the generation of charge carriers leads to increased absorption of the incident THz radiation. While this approach has been proven successful, there are limitations. The control beam wavelength is largely irrelevant as long as the photon energy is above the bandgap energy. Moreover, high peak fluence optical pulses are needed to generate sufficient photocarriers for modulating the THz response.

In recent years, hybrid lead-halide perovskites have been in the spotlight because of their enormous potential for developing high-performance photovoltaics[20, 21]. The appeal arises from a combination of compelling properties that set these materials apart from conventional inorganic semiconductors: (i) a number of solution and vapor-phase deposition techniques can be used to fabricate thin films on a wide range of substrates[22–24], (ii) changes in the chemical structure of the materials allow for tuning of the bandgap over a wide spectral range in the visible[25] and (iii) recent work using methyl-ammonium based perovskites have demonstrated power conversion efficiencies exceeding 20%[26–28].

Compared to three-dimensional (3D) $CH_3NH_3PbI_3$ perovskites, two-dimensional (2D) perovskites self-assemble into alternating organic and inorganic layers that form natural 'multiple-quantum wells', with highly efficient photocurrent[29] and tunable optoelectronic characteristics[30]. Moreover, the interlayer confinement in 2D perovskites gives rise to a large exciton binding energy of order 200 meV[31, 32]. Their absorption spectrum is therefore dominated by a strong exciton band, followed by a slow increase in the absorption above the bandgap energy. These characteristics make the use of 2D perovskites for THz applications promising, although the topic remains largely unexplored[33].

Here we exploit the properties of 2D hybrid lead-trihalide perovskites $–(C_6H_5C_2H_4NH_3)_2PbI_4(PEPI)–$ and its Ruddlesden–Popper (R–P) mixtures with $CH_3NH_3PbI_3$[34], to enable colour selective THz modulation using plasmonic structures fabricated on a semiconductor substrate. In the R–P phase[35], the alternate layers of organic cations and inorganic anion arrange to form an intercalated system of insulating and conducting layers, creating a superlattice structure. By chemically changing the number of the inorganic layers between the adjacent organic layers, the absorption band can be tuned over a wide region of the visible spectral range[26, 36]. When deposited on select semiconductor substrates, we observe enhanced broadband THz absorption, as compared to the bare semiconductor. However, this only occurs when the incident radiation corresponds to the wavelength range associated with exciton absorption.

## Results

### Perovskite synthesis and film growth

The perovskites precursors were synthesized by mixing $PbI_2$, $CH_3NH_3I$ and $C_6H_5C_2H_2NH_3I$ in a dimethylformamide solution with a typical concentration of 0.5 M. The amounts of precursors was

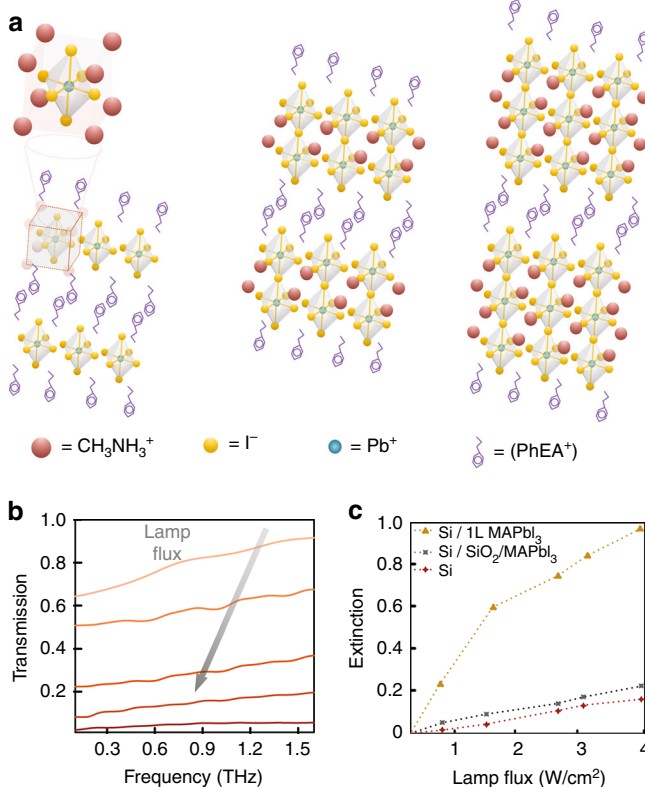

**Fig. 1** 2D PEPI perovskite molecular structure and THz absorption. **a** Molecular arrangement of layered structure of 2D organic–inorganic perovskites for $n = 1$, 2 and 3. In this structure, '$n$' inorganic layers are separated by organic bilayers based on PhEA[+]. **b** THz transmission spectra through an $n = 2$ 2D perovskite deposited onto a high resistivity Si wafer for various photoexcitation intensities, as measured by THz time-domain spectroscopy. The direction of the arrow corresponds to increasing lamp flux. **c** The THz transmission coefficient at 1 THz, measured as a function of the lamp intensity for four different substrates: fused silica/perovskite, bare high resistivity silicon, high resistivity/200 nm $SiO_2$/perovskite, and high resistivity/200 nm $SiO_2$/perovskite. Measurements were performed using optical radiation from a tungsten-halogen lamp, filtered to pass only 360–800 nm

kept at corresponding stoichiometric ratios to form $(C_6H_5C_2H_4NH_3)_2(CH_3NH_3)_{n-1}(PbI_4)_n$ where $n = 1$, 2, 3[32]. Figure 1a shows the typical chemical structure of $n = 1$, 2, and 3 perovskites, where two adjacent layers of octahedral $[PbI_4]^{2-}$ are sandwiched between $[C_6H_5C_2H_4NH_3]^+$, and $[CH_3NH_3]^+$ is embedded in the $[PbI_4]^{2-}$ framework. Thin polycrystalline films, typically ~200 nm thick, were grown via spin casting the precursor solutions onto different substrates which were pre-treated with $O_2$ plasma. The films were annealed subsequently at 100 °C for 30 min and, once cooled, were encapsulated in a layer of poly-methyl methacrylate.

### Photoluminescence and THz transmission measurements

We initially measured the transmitted broadband THz radiation through different samples, while optically exciting the perovskite side of the sample with filtered radiation from a halogen lamp (360–800 nm). In Fig. 1b, we show the optically induced THz transmission spectra measured using ($n = 2$) 2D perovskites deposited directly onto a high resistivity silicon wafer. Care was taken in these measurements to minimize thermal heating artifacts. The THz throughput is seen to decrease approximately uniformly from 0.1 to 1.6 THz with increasing lamp flux. In

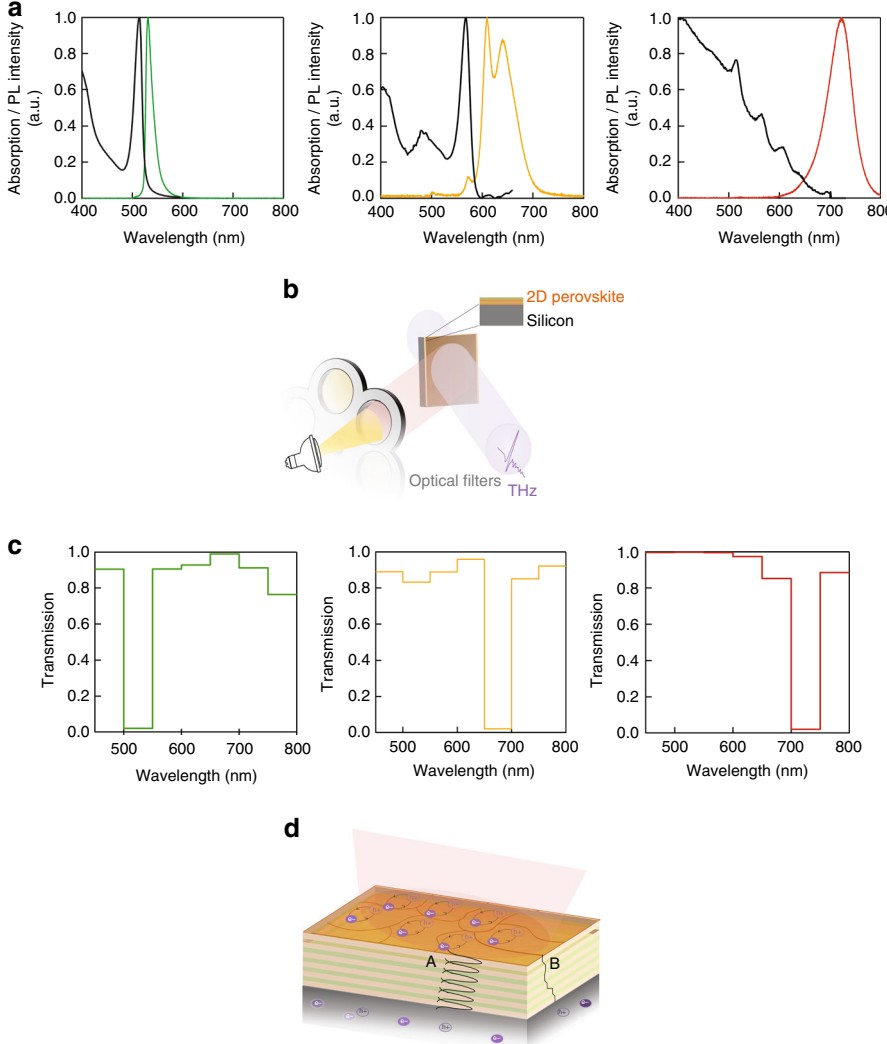

**Fig. 2** Excitation spectra of the THz extinction based on various 2D perovskites. **a** Photoluminescence spectra of the 2D perovskites used in this work and the associated UV–vis absorption (black curves). **b** Schematic depicting the measurement concept. The 2D perovskite samples were photoexcited using a filtered halogen lamp. **c** Transmission spectra of the silicon/perovskite samples for $n = 1$ layer (green), $n = 2$ layer (yellow), and $n = 3$ layer (red) 2D perovskites. The strong absorption of THz radiation occurs in the same narrow-band optical range as the photoluminescence spectra in **a**. **d** An expanded view of the device showing multiple 2D perovskites layers forming a superlattice. The schematic shows two possible mechanisms for exciton dissociation into free carriers. Mechanism A: excitons that are normally confined to the lead-halide plane, exhibit overlapping wavefunctions in the superlattice and tunnel into the silicon substrate, where they can dissociate thereby contributing to the THz absorption. Mechanism B: excitons diffuse through edge states (grain boundaries) where they dissociate into free carriers[38]. These free carriers could diffuse to the underlying silicon through the voids in the polycrystalline films

Fig. 1c, we quantify the THz transmission at 1 THz for perovskite thin layers deposited on different substrates with an optical excitation intensity of 0.5 W/cm². What is clear from the data is the value of placing a perovskite film directly onto a silicon layer (with only a 1 nm native oxide film between the two media). It should be noted that the transmission for $n = 1$ and $n = 3$ 2D perovskite/silicon samples exhibited nearly the same behavior as the $n = 2$ 2D perovskite/silicon sample (Supplementary Note 1). However, when the perovskite layer is placed on a thicker dielectric layer (200 nm), there is no appreciable increase in THz absorption, demonstrating that carrier generation in the perovskite layer itself does not contribute to the THz absorption (Supplementary Note 2). It is the interface between the silicon and perovskite that dramatically enhances broadband THz absorption upon optical illumination. It is worth noting that graphene/semiconductor bilayer structures have been shown to exhibit enhanced photo-induced absorption, mediated by charge transfer at the interface. This may yield insight into the enhanced

absorption mechanism in our samples[37]. However, a detailed understanding of the carrier dynamics at this interface are beyond the scope of this study, although further examination may be beneficial not only for this application, but also for the development of photovoltaic devices.

We synthesized three R–P phase perovskite variants, $n = 1$ to $n = 3$, that exhibit various bandgaps between 2.5 eV for $n = 1$ to 1.8 eV for $n = 3$[31]. In Fig. 2a, we show the absorption and photoluminescence emission spectra for each $n$-variant. These PL spectra show a slight red-shift compared with the absorption bandedges. This occurs because the photoexcitations preferentially relax to the impurities of larger $n$-number (although they are only of a trace amount). This agrees also with published data[31, 32], demonstrating both the quality and distinctness of each material. To better understand the excitation wavelength dependence on the THz absorption in the 2D perovskites/Si samples, we optically illuminated each sample using a series of 50 nm wide bandpass filters ranging from 450 to 800 nm. A

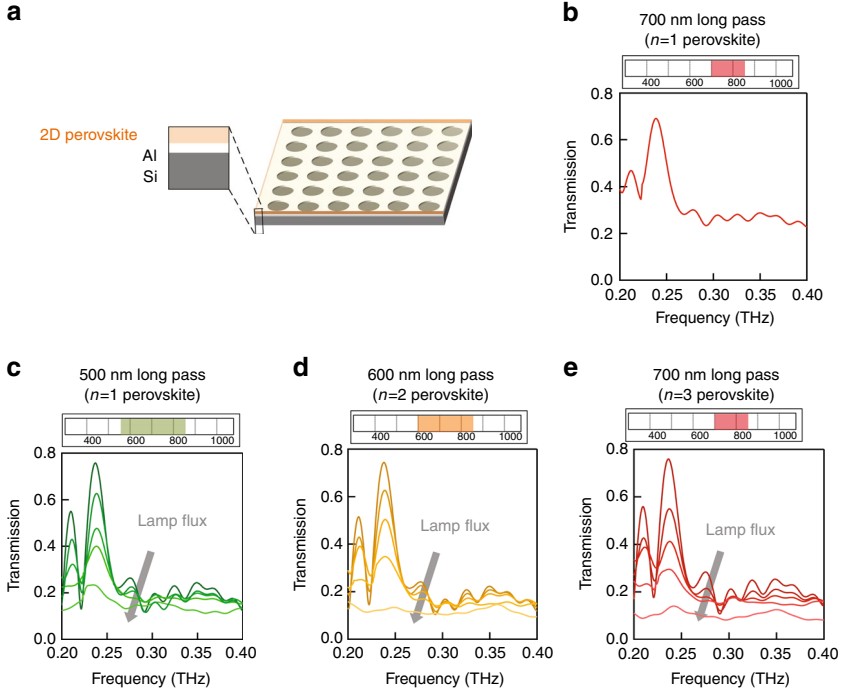

**Fig. 3** Attenuation of THz plasmonic resonances. **a** Schematic diagram of the device composed of a high resistivity silicon substrate with a 200 nm perforated aluminum film topped by a 200 nm thick 2D perovskite thin film. **b** Transmission spectrum of the aperture array demonstrating negligible attenuation for the $n = 1$ perovskite/Si sample when excited with 700 nm long pass filter (i.e., no excitons are generated). **c**–**e** Optical excitation induced attenuation of the transmitted THz transmission through the metallic hole array. The dips on the high frequency side of the two lowest order resonances occur at . These arrays were excited using 500 nm long pass, 600 nm long pass, and 700 nm long pass filters, respectively. The optical excitation range is shown above each spectrum. Nearly 100% attenuation of the plasmonic resonances was achieved as the lamp flux was increased (the arrow direction corresponds to increasing lamp flux)

schematic diagram of the experimental setup is shown in Fig. 2b, with the corresponding data shown in Fig. 2c.

Interestingly, THz absorption in these samples increases appreciably only when the wavelength of the incident optical radiation corresponds to the wavelength of the exciton resonance, but not at higher energies that corresponds to band-to-band transition. To understand this phenomenon, we consider the sample structure. When a thin perovskite film is deposited onto a substrate, it consists of a stack of 2D R–P layers, each of which naturally forms a quantum well. As shown schematically in Fig. 2d, the resulting structure forms a superlattice, in which the barriers are thin, such that the exciton wavefunctions in adjacent wells strongly overlap. Upon illumination with above bandgap optical excitation, both free carriers and excitons are photo-generated in each layer. In the case of isolated layers, the photoexcitations are usually confined to their respective lead-halide planes[36]. However, in a superlattice structure the photoexcitations may be delocalized, which allows for tunneling between layers. The optically induced THz absorption data shown in Fig. 2c suggests that only excitons can tunnel vertically through the superlattice and into the silicon substrate. There, they can dissociate and add to the overall carrier density in the Si substrate (Supplementary Fig. 2), thereby substantially increasing the THz absorption. Free carriers, on the other hand, do not appear to be able to tunnel through the multilayers and into the silicon substrate. Thus, only excitation wavelengths causing exciton formation lead to increased THz absorption. This also explains why we do not observe any THz absorption when a 2D perovskite film is deposited onto a dielectric (Supplementary Fig. 2). Blancon and co-workers recently showed that excitons in 2D perovskites can diffuse to lower energy edge states where they can dissociate into free carriers[38]. This may be an alternate explanation for

increased carrier generation in the silicon substrate, where excitons are able to reach the substrate via voids between grain boundaries. Finally, we note that excitons do not dominate the absorption in 3D MAPbI$_3$ perovskites and, thus, we do not observe an analogous wavelength (colour) selective control of the THz absorption. Instead, 3D perovskites behave much the same as conventional semiconductors for this specific application, as discussed in Supplementary Notes 3 and 4.

**Active THz devices.** This discovery can be used to create optical wavelength (colour) selective THz modulators. Figure 3 summarizes the basic idea and experimental results. We fabricated a series of periodic subwavelength aperture arrays[39, 40] in 200 nm thick aluminum thin films deposited on high resistivity silicon wafers. Subsequently, a $n = 1$, $n = 2$, or $n = 3$ hybrid perovskite thin film was deposited on top of the Al structure, so that it was in direct contact with the silicon only inside of each aperture, as shown schematically in Fig. 3a. Figure 3b shows that the $n = 1$ sample exhibits no optical intensity dependence when illuminated at 700 nm (Fig. 2c), since that wavelength does not correspond to photogenerated excitons in the perovskite layer. The transmission spectra are also measured as a function of the optical intensity and shown in Fig. 3c ($n = 1$ perovskite) through Fig. 3e ($n = 3$ perovskite), when illuminated using narrow-band radiation at the optimal wavelengths determined in Fig. 2. In each case, with increasing optical intensity, the increased THz absorption within the apertures leads to lower overall THz transmission. These data further demonstrate that the observations are associated with increased exciton photogeneration rather than due to local heating or other artifacts.

A significant advantage of perovskites over other conventional crystalline semiconductors is that thin films can be fabricated

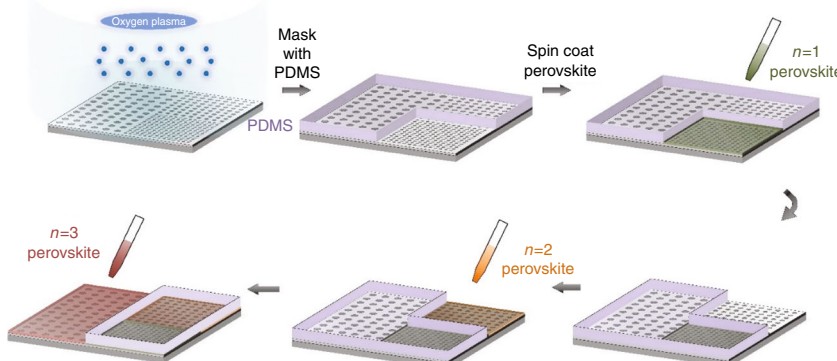

**Fig. 4** Schematic showing solution-based fabrication of colour selective THz modulator. (clockwise) The solution processed 2D perovskites having different *n*, are spin coated selectively onto the different sections of the aperture array-based structure using a series of PDMS masks

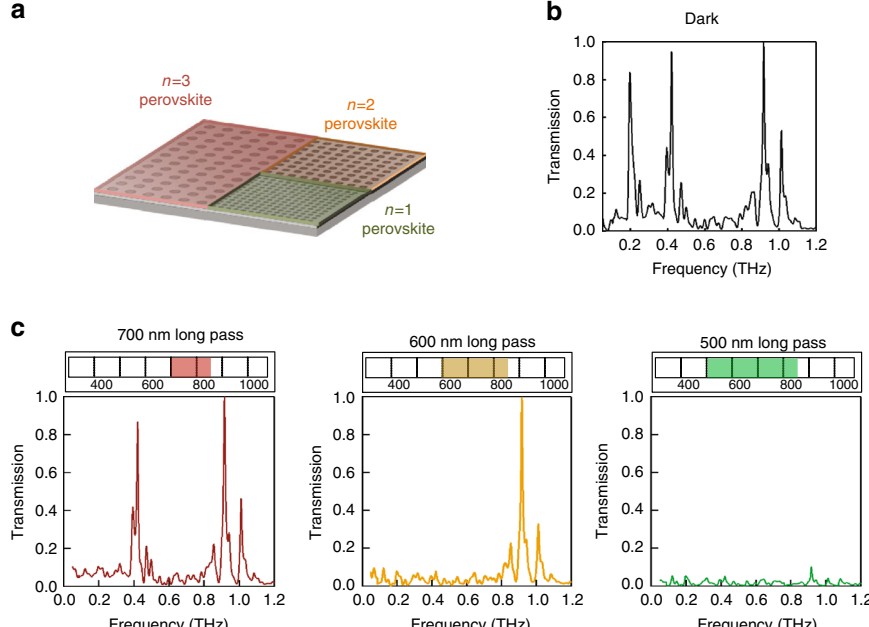

**Fig. 5** Excitation selective THz modulator. **a** Schematic of a resonant device that consists of three different aperture array sections coated with different 2D perovskite layers as denoted. **b** The THz spectrum of the device in the dark showing the resonant response for all three sections. The lowest order resonances for each section have peaks at 0.22 THz, 0.41 THz, and 0.95 THz, respectively, that correspond directly to the aperture spacing in each section. The smaller bands next to the primary bands correspond to higher order plasmonic resonances. **c** (left to right) THz spectra of the device illuminated with 700 nm long pass filter, 600 nm long pass filter, and 500 nm long pass filter, respectively. The optical excitation range is shown above each spectrum. For example, a 600 nm long pass filter allows for illumination from 600 to 850 nm. Thus, a broader optical spectrum can span multiple exciton excitation ranges. The entire device was uniformly illuminated in each case

using comparatively simple solution-based and vapor-phase deposition techniques[22–24]. Thus, multiple perovskites can be deposited on different regions of the same device. To demonstrate the efficacy of this approach, we fabricated another plasmonic structure in which three separate subwavelength aperture arrays with different periodicities, though with identical aperture radius to aperture spacing ratios, were placed adjacent to one another, as shown schematically in Fig. 4. We designed the number of apertures in each section to ensure that the magnitude of the measured THz amplitude for the lowest order resonance was approximately identical, regardless of frequency. We then deposited 2D perovskite thin films with different *n* values on top of each region. The basic fabrication steps, shown in Fig. 4, utilized a series of poly-dimethyl siloxane (PDMS) shadow masks to define the deposition regions. The resulting array sections had

lowest order resonance peaks at ~ 0.22 THz (*n* = 3), ~ 0.41 THz (*n* = 2), and ~ 0.95 THz (*n* = 1).

Figure 5 summarizes the experimental results that demonstrate the utility of colour selective modulation of THz radiation. The transmission spectrum associated with the device is shown in Fig. 5a and consists of three bands of resonances, one for each aperture array region. Each of these bands consists of multiple resonances that can be related directly to the periodicity and dielectric properties of the adjacent materials (perovskites and silicon)[40]. In contrast to the earlier demonstrations discussed above, we now use long pass optical filters to progressively reduce and ultimately erase multiple bands of resonances. Using the experimental geometry shown in Fig. 2b with a 700 nm long pass filter, the resonances in the lowest frequency band of resonances (near 0.2 THz) can be reduced in amplitude (Supplementary

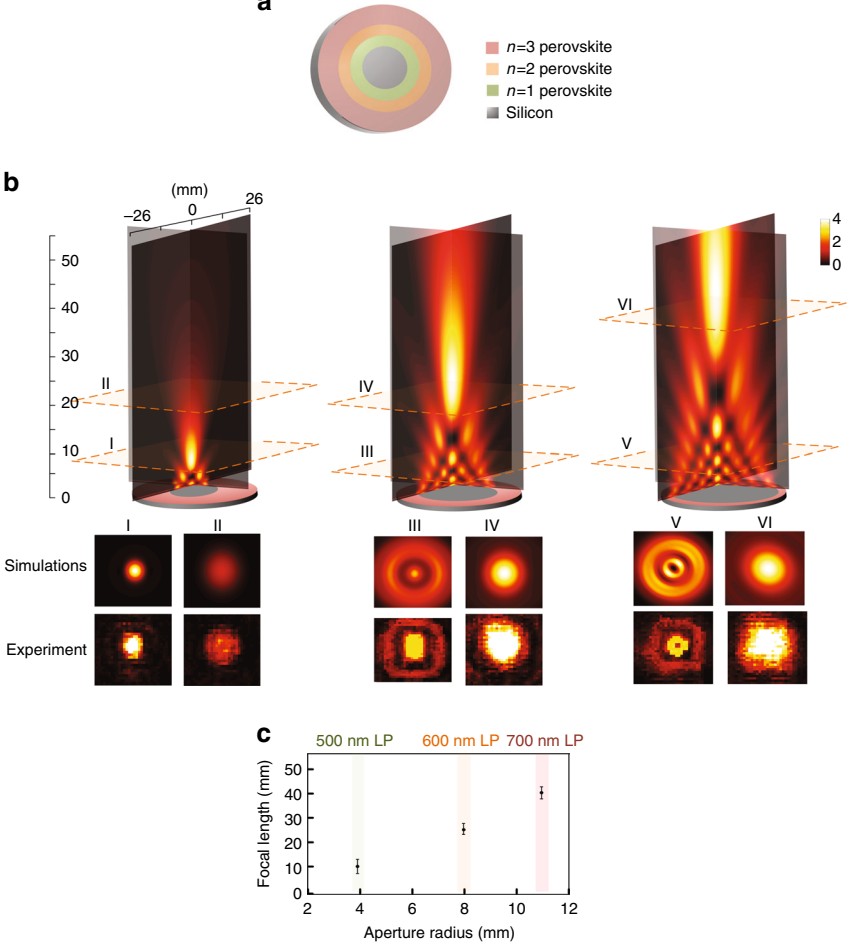

**Fig. 6** Excitation tunable THz focusing element. **a** Sample schematic, where concentric rings of $n = 1, 2, 3$ perovskite films were deposited such that upon selective excitation, the structure allows change of dielectric (unexcited) aperture size in metallic (photoexcited) rings; **b** (left to right) The FDTD simulation results for the focused rays through the different apertures (4, 8, and 11 mm). Panels I–VI show experimental THz images measured at positions of cross-sectional planes as denoted, as well as the corresponding cross-sections from numerical simulations. **c** The aperture radius and focal length at 500 nm long pass, 600 nm long pass, and 700 nm long pass excitation. Error bars, s.d. for repeated measurements (eight in total)

Note 5) in a controlled manner by varying the lamp flux. For the maximum optical intensity used here, this band of resonances disappear completely (Fig. 5b). However, the higher frequency resonances remain unchanged, because only the region covered by the $n = 3$ perovskite is affected. When a 600 nm long pass filter is used, only the resonance bands near 0.2 and 0.4 THz, corresponding to the $n = 3$ and $n = 2$ perovskites undergo modulation (Fig. 5c). Finally, when a 500 nm long pass filter is used (Fig. 5d), all three sets of resonances are impacted.

The perovskite/Si structure opens numerous possibilities for other device implementations. As an example, we demonstrate the ability to optical control the focusing of THz radiation. The specific implementation relies on creating a circular aperture in which the diameter can be altered in a colour selective manner. For a circular aperture, the intensity variation along the axis is determined by the incident wavelength, aperture size, and distance from the aperture, and can be described analytically by Rayleigh-Sommerfeld diffraction[41, 42]. We patterned a silicon substrate with cylindrical concentric rings $n = 1, 2,$ and 3 perovskites, as shown schematically in Fig. 6a. Using the same illumination sequence as in Fig. 5 with a 100 GHz narrow-band frequency THz source, we were able to change the effective aperture radius from 11 mm (700 nm long pass), to 8 mm (600 nm long pass) to 4 mm (500 nm long pass). In Fig. 6b, we show the numerically simulated THz spatial pattern for the three

apertures mentioned above, along with cross-sections in the designated planes using both experimental THz imaging measurements and numerical simulations. The measured and simulated cross-sections agree well and correspond to focal lengths of 10 mm (11 mm radius aperture), 25 mm (8 mm radius aperture), and 40 mm (4 mm radius aperture), as shown in Fig. 6c.

## Discussion

In summary, we have demonstrated that 2D hybrid organic–inorganic lead-trihalide perovskites offer a number of unique characteristics for fabricating active THz devices that distinguish them from structures fabricated using conventional semiconductors. Notably, the bandgaps of these materials, as well as the associated exciton absorption band, can be easily altered. Moreover, these perovskites can be deposited using a variety of conventional thin film deposition techniques and they yield a photoresponse that is dramatically larger than the bare semiconductor onto which they are deposited. Using a plasmonic structure over which multiple perovskites are deposited, we have shown near complete suppression of transmission resonances using only relatively low intensity narrow-band excitation from a halogen lamp. Thus, this approach overcomes the limitations encountered with existing modulators: the devices require only relatively simple fabrication techniques[3–12] and when compared

to optically induced modulation, THz absorption now depends upon the excitation optical wavelength and a halogen lamp can be used instead of an ultrafast laser source. These materials offer new possibilities for greater control over active THz devices.

## Methods

**Fluorescence and UV–visible measurements.** Fluorescence spectra were taken with an optical fiber based spectrometer (Ocean Optics USB4000ES) by exciting the films on glass substrates with a 447 nm semiconductor laser (50 mW). A 500 nm long pass filter was used before the spectrometer to block the residual laser beam. UV–visible absorption spectra were carried out with Agilent Cary UV–vis Spectrophotometer (200 nm to 2.5 μ).

**FTIR measurements.** Photo-induced free carrier absorption was characterized using an FTIR (Fourier transform infrared) spectrometer in various samples. A 2.8 eV (442 nm) cw diode laser was used as the optical pump and the IR probe beam was provided by the FTIR. The 2D perovskite films were deposited on Si and KBr substrates and placed in a cryostat in which the temperature could be varied from 45 to 300 K. The pump and probe beams were co-incident on the sample films, and the transmission, $T$, and change of the transmitted probe beam ($\Delta T$) was detected by a DLaTGS detector. The photoabsorption spectrum was calculated from measurements of the fractional change in transmission ($\Delta T/T$) using approximately 6000 scans of the FTIR spectrometer with the pump beam on and off.

**Multi-perovskite device fabrication.** The patterned multi-perovskite structures were prepared using a step-by-step spin-coating method. A silicon wafer with thin film aluminum aperture arrays were treated with $O_2$ plasma to obtain a hydrophilic surface. A thin layer of PDMS mask was then attached onto the silicon wafer before spin-coating the perovskite precursor on top. After annealing the wafer to remove the residual solvent, the uncovered part was uniformly coated with perovskite material and the PDMS mask was removed. By repeating this procedure multiple times, the silicon wafer surface was patterned with multiple 2D perovskite compounds. Finally, the prepared sample was annealed at 100 °C for 1 h to obtain polycrystalline perovskite films. The sharpness and size of the patterns were determined by the PDMS mask, with a spatial resolution of ~1 mm.

**THz transmission and imaging measurements.** The transmission spectra were obtained using standard THz time-domain spectroscopy (THz TDS) measurements[43]. We used nonlinear optical crystals for both THz generation and detection stimulated by an ultrafast Ti:sapphire laser[44]. The broadband, linearly polarized THz radiation generated by the emitter was collected and collimated by an off-axis parabolic, such that the radiation was normally incident on the samples. A second off-axis parabolic mirror placed after the filter was used to focus the transmitted THz radiation onto the detection crystal for measurement via electro-optic sampling. For THz imaging, we used a narrow-band polarized electronic THz source operating at 0.1 THz. The transmitted radiation was imaged using a THz focal plane array.

**Data availability.** The data that support the findings of this study are available from the corresponding authors upon reasonable requests.

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

### Acknowledgements

This work was supported by the NSF MRSEC program at the University of Utah under grant #DMR 1121252. Support from the NSF ECCS 1607516 for sample preparation is also acknowledged.

### Author contributions

All authors contributed to all aspects of this work.

### Additional information

**Competing interests:** The authors declare no competing financial interests

