## [Peer Review File · Nature Communications]

Reviewers' comments:

Reviewer #1 (Remarks to the Author):

The authors report a new terahertz wave modulator that modulates terahertz radiation over a wide frequency band at a high modulation depth. Moreover, the approach allows frequency-selective modulation of terahertz waves due to resonant band edge exciton formation. They explain the mechanism of the narrowband modulation capability and carry out measurements to support their theoretical explanation. As a direct application, they successfully experimentally demonstrate tunable focusing by use of an optically reconfigurable circular aperture and corroborate their findings by numerical simulations.

In my opinion, the paper perfectly matches the scope of the journal. Furthermore, the findings are very interesting and well supported by experiments and theoretical considerations. The new approach hosts a high potential for new applications in terahertz research. The paper is well organized and understandable to a broad readership. For this reason, I recommend the paper for publication, but suggest some (minor) revisions:

1. The authors clearly demonstrate that 2D hybrid organic inorganic lead-trihalide perovskites on a high resistivity wafer can act as an optically tunable modulator for terahertz waves over a wide frequency band. They do not explain the physical mechanism behind and argue that this would be out of the scope of the article. I agree with that. Indeed, I think that the topic is not trivial at all. The authors assume that interface effects lead to tunneling of free carriers into the silicon substrate. This is supported by the observation that deposition of the 2-D materials on top of dielectrics does not offer the possibility to modulate terahertz waves. In fact, a similar effect has been observed by other authors in a system that consists of a graphene layer on top of high resistivity silicon. They also demonstrated wide-band terahertz wave modulation at a high modulation depth by optical tuning of the graphene layer. Also in this case, optical tuning could not be observed in graphene on BCB structures and the reason for modulation in graphene on silicon structures is similar to the mechanism in the 2-D perovskite on silicon structures. Therefore, I would suggest that the authors cite the paper "Spectrally Wide-Band Terahertz Wave Modulator Based on Optically Tuned Graphene" by P. Weis et al., ACS Nano 6, 9118 (2012). Although I am not completely sure whether the explanation in the ACS Nano paper is appropriate for the case of 2-D perovskite on top of silicon, it might be worth for the authors to take a look at it.
2. Is there an explanation, why excitons can tunnel through the superlattice and free carriers cannot? Is it due to the spatial distribution of the wavefunction? Maybe the authors can clarify that.
3. In the Supplementary Material, grammatical errors are in line 60 and in line 75.

Reviewer #2 (Remarks to the Author):

The authors present an innovative approach for optically controlling the propagation and attenuation properties of THz wave by using 2D hybrid organic inorganic lead-trihalide perovskites. They demonstrate that THz absorption depends on the excitation optical wavelength and a halogen lamp has been used instead of a laser source. It is really an interesting work and the manuscript can be published in the Nature Communications providing the following question or suggestion can be solved.

1. For the experimental results showing in figure 5c and 5d, instead of modulating one THz peak as 700nm light did in around 0.2 THz, why the exciting light with 600nm wavelength can modulates both 0.2 THz and 0.4THz and the light with 500nm wavelength can modulates all three THz transmission peaks?

2. I suggest that the inset of figure 3b be shown as a separate figure to show the exciting light intensity dependence for the transmission THz wave.
3. The caption for figure 6 does not match the pictures there.

Reviewer #3 (Remarks to the Author):

Chanana et al describe a THz modulator based on 2D perovskites thin films deposited on silicon. Their work is well presented and will be of interest to both the THz and perovskite communities. Below are a few comments small comments which, when addressed, will make the paper stronger.

It would be helpful to see UV-vis data for all of the samples investigated. The UV-vis spectra of these materials are very sensitive to impurities and could therefore strengthen the authors' explanation of the red-shifted PL spectra.

The authors state that the operative mechanism in their device involves excitons tunneling between layers of the 2D perovskite.

As the alignment of these materials often depends on fabrication method and the n value (higher n is often more disordered), it would be useful if they could comment of the degree of alignment that they expect in their samples and how this would affect the efficiency of their device. Better yet would be if they could support their argument with GIWAXS data or something similar.

Finally, the authors list a large number of previous THz modulators in the literature (references 3-19) but do not comment about how their device compares. It would be helpful if they could give statistics about efficiency and comment about how they compare to similar devices.

Reviewer #1 (Remarks to the Author):

We thank the reviewer for the positive comments regarding the manuscript.

Comment #1: *The authors clearly demonstrate that 2D hybrid organic inorganic lead-trihalide perovskites on a high resistivity wafer can act as an optically tunable modulator for terahertz waves over a wide frequency band. They do not explain the physical mechanism behind and argue that this would be out of the scope of the article. I agree with that. Indeed, I think that the topic is not trivial at all. The authors assume that interface effects lead to tunneling of free carriers into the silicon substrate. This is supported by the observation that deposition of the 2-D materials on top of dielectrics does not offer the possibility to modulate terahertz waves. In fact, a similar effect has been observed by other authors in a system that consists of a graphene layer on top of high resistivity silicon. They also demonstrated wide-band terahertz wave modulation at a high modulation depth by optical tuning of the graphene layer. Also in this case, optical tuning could not be observed in graphene on BCB structures and the reason for modulation in graphene on silicon structures is similar to the mechanism in the 2-D perovskite on silicon structures. Therefore, I would suggest that the authors cite the paper “Spectrally Wide-Band Terahertz Wave Modulator Based on Optically Tuned Graphene” by P. Weis et al., ACS Nano 6, 9118 (2012). Although I am not completely sure whether the explanation in the ACS Nano paper is appropriate for the case of 2-D- perovskite on top of silicon, it might be worth for the authors to take a look at it.*

Response #1: We appreciate the comment and pointing out the article by P. Weis *et al.*, which we have added the reference list. We have also added the following text on page 5 of the manuscript: “It is worth noting that graphene/semiconductor bilayer structures have been shown to exhibit enhanced photo-induced absorption, mediated by charge transfer at the interface. This may yield insight into the enhanced absorption mechanism in our samples [37].”

In that publication, the authors attribute the enhanced conductivity to diffusion of carriers from silicon to high mobility graphene, which gives rise to broadband absorption of THz radiation. However, it is not clear that the same phenomenon is responsible for enhanced THz absorption for 2D perovskite/silicon samples. This is based on the two observations:

- a. The 2D and 3D perovskite films are known to have relatively poor charge carrier mobilities. Even in the single crystals have been reported to have modest values of up to $100 \text{ cm}^2/\text{V}\cdot\text{sec}$.
- b. The excitation dependent response of these perovskite films showed enhanced absorption in samples only when photoexcited with corresponding optical frequencies (Fig 2: Excitation spectra of the THz extinction based on various 2D perovskites). This observation suggests charge transfer of photoexcited carriers from perovskite to silicon.

In response to the reviewer’s concern, we have measured the photoinduced free carrier absorption (FCA) in a variety of different samples, such as: 2D perovskite on silicon, 2D perovskite on KBr (dielectric) substrate, and bare Si from 17 – 80 THz using FTIR spectroscopy. Interestingly, 2D perovskite on Si shows same characteristic FCA response as that in bare Si. This contrasts with 2D perovskite on KBr, which shows non-Drude like behavior. Such carrier dynamics further indicates that charges reached in the silicon substrate, in the 2D perovskite/ silicon bilayer structure. *We have created a new Section 3 in the Supplementary Information that describe these results.*

Comment #2: *Is there an explanation, why excitons can tunnel through the superlattice and free carriers cannot? Is it due to the spatial distribution of the wavefunction? Maybe the authors can clarify that.*

Response #2: We suspect that defects and dangling bonds at the grain boundaries give rise to significantly enhanced recombination for free carriers, leaving excitons as primary means for enhancing the conductivity. In the original manuscript, we pointed out two potential mechanisms for exciton tunneling from the 2D perovskites layer to the silicon substrate: either via exciton wavefunction overlap in a perovskites-based superlattice structure, or edge states in the layered perovskites (the latter mechanism is discussed in the recent *Science* paper by J-C. Blancon et al. [ref. 38 in the manuscript]. We

have modified Figure 2d and the corresponding caption to more clearly show these two possible mechanisms.

Comment #3: In the Supplementary Material, grammatical errors are in line 60 and in line 75.

Response #3: We thank the reviewer for pointing out the error and have corrected it.

Reviewer #2 (Remarks to the Author):

We thank the reviewer for the positive comments regarding the manuscript.

Comment #1: For the experimental results showing in figure 5c and 5d, instead of modulating one THz peak as 700nm light did in around 0.2 THz, why the exciting light with 600nm wavelength can modulates both 0.2 THz and 0.4THz and the light with 500nm wavelength can modulates all three THz transmission peaks?

Response #1: We apologize for the confusion. When a 700 nm long pass filter is used, photon absorption occurs only for excitons in the n=3 layered perovskite and, therefore, only the resonances associated with the aperture hole underneath the n=3 perovskite will be suppressed. In contrast, when a 600 nm long pass filter is used, photon absorption occurs for excitons in both the n=2 and n=3 layered perovskites, since the optical excitation is no longer a narrow bandwidth that covers only one type of 2D perovskites.

We could have alternatively used only narrowband excitation to suppress any one set of resonances, but instead chose to use progressively larger bandwidths that cover one, two and finally all three layered perovskites resonances. *In the revised manuscript we have added images above each spectrum in Figure 2 that more clearly shows the excitation band used.* Note that an IR filter removes all optical radiation above 800 nm.

Comment #2: I suggest that the inset of figure 3b be shown as a separate figure to show the exciting light intensity dependence for the transmission THz wave.

Response #2: We have modified the figure as requested.

Comment #3: The caption for figure 6 does not match the pictures there.

Response #3: We have corrected the caption for Figure 6.

Reviewer #3 (Remarks to the Author):

We thank the reviewer for the positive comments regarding the manuscript.

Comment #1: It would be helpful to see UV-vis data for all of the samples investigated. The UV-vis spectra of these materials are very sensitive to impurities and could therefore strengthen the authors' explanation of the red-shifted PL spectra.

Response #1: We appreciate reviewer's comment and have modified Figure 2 to include UV-Vis data for the samples along with the PL spectra. Indeed the PL red shift may be the result of exciton migration into lower energy states in the 2D perovskite film.

Comment #2: The authors state that the operative mechanism in their device involves excitons tunneling between layers of the 2D perovskite. As the alignment of these materials often depends on fabrication method and the n value (higher n is often more disordered), it would be useful if they could comment of the degree of alignment that they expect in their samples and how this would affect the efficiency of their

device. Better yet would be if they could support their argument with GIWAXS data or something similar.

Response #2: In order to effectively comment on the relationship between the degree of alignment and the device performance, we would need to test a large number of samples in which there is variation in the film properties. This is an interesting proposal that suggests future work. However, we believe that it is beyond the scope of the present work.

***Comment #3:** Finally, the authors list a large number of previous THz modulators in the literature (references 3-19) but do not comment about how their device compares. It would be helpful if they could give statistics about efficiency and comment about how they compare to similar devices.*

Response #3: We appreciate the reviewer's comment. As the reviewer has pointed out, there have been a number of different THz modulators. Our works stands out in two ways:

1. We achieve nearly 100% transmission suppression using only a halogen lamp. All of the cited papers used (amplified) ultrafast lasers.
2. We demonstrate optical-wavelength dependent changes in the transmission. This has not been shown previously

We have modified one sentence in the conclusion from “Thus, this approach overcomes the two limitations commonly encountered with conventional semiconductors: THz absorption now depends upon the excitation optical wavelength and a halogen lamp can be used instead of an ultrafast laser source.” To “Thus, this approach overcomes the limitations encountered with existing modulators: the devices require only relatively simple fabrication techniques [3-12] and when compared to optically induced modulation, THz absorption now depends upon the excitation optical wavelength and a halogen lamp can be used instead of an ultrafast laser source.”

REVIEWERS' COMMENTS:

Reviewer #2 (Remarks to the Author):

The authors have answered the question and revised the manuscript according to the reviewer's suggestions. I agree that the manuscript be accepted by Nature Communications for publication.

Reviewer #3 (Remarks to the Author):

I am satisfied with the authors' responses and have no further comments.